# A novel hypergraph model for identifying and prioritizing personalized drivers in cancer

**Naiqian Zhang**[1☯]*, **Fubin Ma**[1☯], **Dong Guo**[1,2☯], **Yuxuan Pang**[3], **Chenye Wang**[1], **Yusen Zhang**[1], **Xiaoqi Zheng**[4], **Mingyi Wang**[1,2]*

1 School of Mathematics and Statistics, Shandong University, Weihai, China, 2 Department of Central Lab, Weihai Municipal Hospital, Shandong University, Weihai, China, 3 SDU-ANU Joint Science College, Shandong University, Weihai, China, 4 Center for Single-Cell Omics, School of Public Health, Shanghai Jiao Tong University School of Medicine, Shanghai, China

☯ These authors contributed equally to this work.
* nqzhang@email.sdu.edu.cn (NZ); wangmingyi1973@outlook.com (MW)

## Abstract

Cancer development is driven by an accumulation of a small number of driver genetic mutations that confer the selective growth advantage to the cell, while most passenger mutations do not contribute to tumor progression. The identification of these driver genes responsible for tumorigenesis is a crucial step in designing effective cancer treatments. Although many computational methods have been developed with this purpose, the majority of existing methods solely provided a single driver gene list for the entire cohort of patients, ignoring the high heterogeneity of driver events across patients. It remains challenging to identify the personalized driver genes. Here, we propose a novel method (PDRWH), which aims to prioritize the mutated genes of a single patient based on their impact on the abnormal expression of downstream genes across a group of patients who share the co-mutation genes and similar gene expression profiles. The wide experimental results on 16 cancer datasets from TCGA showed that PDRWH excels in identifying known general driver genes and tumor-specific drivers. In the comparative testing across five cancer types, PDRWH outperformed existing individual-level methods as well as cohort-level methods. Our results also demonstrated that PDRWH could identify both common and rare drivers. The personalized driver profiles could improve tumor stratification, providing new insights into understanding tumor heterogeneity and taking a further step toward personalized treatment. We also validated one of our predicted novel personalized driver genes on tumor cell proliferation by vitro cell-based assays, the promoting effect of the high expression of Low-density lipoprotein receptor-related protein 1 (*LRP1*) on tumor cell proliferation.

## Author summary

In this study, using the TCGA dataset studies as benchmark datasets, we explored the application of the commonality among patients of the same cancer type in personalized driver gene prediction. We proposed a hypergraph model and a generalized random walk method to rank the mutated genes of a patient based on their impact on the abnormal expression of downstream genes in a group of samples rather than an individual sample.

**Data Availability Statement:** The source code and data used to generate the results presented in this manuscript are available from the GitHub repository: https://github.com/ShandongUniversityMasterMa/PDRWH.

**Funding:** This work has been supported by the National Natural Science Foundation of China [62072277 to NZ, 61972257 to XZ and 61877064 to YZ]. The funders had no role in study design, data collection and analysis, decision to publish, or preparation of the manuscript.

Following the extensive experimental results on 16 cancer datasets and the comparative analysis across five cancer types, we have observed that the PDRWH method exhibits remarkable effectiveness in identifying known general driver genes and tumor-specific driver genes. In a few words, our method can provide a more accurate personalized catalog of driver mutations for each patient, and the predicted personalized driver genes can be applied to improve tumor stratification. It can also provide oncologists with a reliable candidate gene list to assist treatment decisions, thus potentially promoting the development of personalized medicine.

## Introduction

Cancer is a collection of genetic diseases characterized by abnormal and uncontrolled cellular growth, which are primarily caused by the accumulation of genetic alterations [1–3]. Previous evidence has shown that a small fraction of genomic and transcriptomic altered genes, called cancer driver genes, could modify transcriptional programs and result in abnormal cell proliferation and eventually tumorigenesis [4–6]. The majority of detected altered genes are passengers that do not contribute to the oncogenic process. Distinguishing cancer driver genes from numerous functionally neutral passenger mutation genes is critical for providing clinically characterized insights into tumor biology. And it has led to the development of a paradigm of targeted anticancer therapies, and the search for biomarkers of prognosis and response to cancer treatments [7–9].

With recent advances in genomics technologies, comprehensive platforms such as the Cancer Genome Atlas (TCGA), have led to the characterization of the molecular signatures of human cancers spanning 33 cancer types, providing an unprecedented opportunity to develop computational methods for driver gene identification [10]. Many bioinformatics tools are dedicated to identifying driver mutations from passengers in a cohort of patients [11–15]. However, since cancer patients possess different genomes and their disease might be driven by different driver genes [16, 17], it is necessary to investigate personalized cancer drivers specific to an individual patient. For example, DawnRank applies the PageRank algorithm to evaluate the impact of genes on the overall differential expression of its downstream genes in a molecular interaction network [18]. Prodigy prioritizes candidate personalized driver genes by quantifying the impact of mutated genes on deregulated pathway based on the patient's tumor mutation and expression profiles [19]. According to a single-sample network control strategy, Guo et al. developed SCS to detect the minimum set of driver nodes that could achieve the maximal coverage of individual differentially expressed genes during the transition from the normal state to the disease state [20]. Despite achieving promising results, these personalized driver prioritization methods take into account the data available from a single sample to produce a ranking of drivers for every specific patient, neglecting the availability of data from other samples. More importantly, they are overly dependent on the data quality of individual samples, with poor tolerance for noise and low reliability of the results. To address this issue, PersonaDrive aims to utilize the comprehensive whole cohort data for guiding the personalized driver prediction [21]. This is achieved by constructing a bipartite graph to model pairwise relationships among the set of mutated genes and the differently expressed genes. However, it is widely acknowledged that the bipartite graph, as a type of simple graph, is limited to capturing pair-wise relationships between nodes and cannot represent more complex relationships. An important aspect of cancer that has been overlooked by existing methods is that patients with the same driver gene mutations are likely to share the same carcinogenic

mechanism, unlike patients with only the same passenger mutations [22, 23]. Taking this into account, incorporating higher-order relationships between mutated genes and abnormally expressed genes in computational models holds significant promise for improving personalized cancer driver prediction.

In this study, we present a novel method named PDRWH (Prioritizing Personalized Cancer Driver Genes via Random Walks on a Hypergraph), inspired by the effectiveness of hypergraphs in modeling biology networks, data structures, and other systems [24–27]. Unlike methods that rely on data from a specific sample, PDRWH integrates data from a cohort to generate personalized driver gene predictions, enabling a more comprehensive analysis of the collective information across multiple samples. Under the assumption that the impact of a potential driver gene can be determined by its effect on the genes regulated by it, PDRWH ranks potential driver genes based on the influence of mutated genes on transcriptional networks across the cohort of samples. To achieve this, a hypergraph model is proposed to effectively represent high-order relationships among genes. It captures the implicit intrinsic regulatory associations among genes within each sample by connecting a large number of mutated genes and aberrantly expressed genes in the corresponding hyperedge. Additionally, this model accurately characterizes the coexistence of mutated genes and aberrantly expressed genes across diverse samples. PDRWH quantifies the impact of each mutated gene across the group of samples by performing a generalized random walks algorithm on the personalized hypergraph. Evaluated across datasets from 16 cancer types in TCGA and benchmarked against existing driver gene prediction methods using five cancer type datasets, PDRWH consistently demonstrates superior performance in identifying both known general driver genes and tumor-specific driver genes. Notably, PDRWH excels at simultaneously identifying both common and rare driver genes. The predicted personalized driver gene profiles can not only improve tumor stratification but also provide oncologists with a reliable candidate gene list to assist in treatment decisions. To validate the effectiveness of PDRWH, we experimentally verified a predicted personalized driver gene *LRP1* through in vitro cell assays.

## Results

### An overview of PDRWH

The PDRWH method is a novel integrated genome/transcriptome analysis approach designed to identify candidate personalized driver genes by leveraging the influence of mutated genes on biological networks across cohort samples. We hypothesize that samples with the same cancer types of cancer display higher similarity in molecular characteristics and disease mechanisms compared to samples with different types of cancer. Leveraging samples from the same cancer type, specifically those closely resembling the target sample, has the potential to improve the personalized driver genes prediction. A schematic overview of PDRWH is illustrated in **Fig 1**. PDRWH requires knowledge of cohort samples from a specific cancer type in TCGA, including the somatic mutation and the gene expression profiles, as well as a gene interaction network (**Fig 1A**). To ensure uniform baseline values and ranges for all genes across the cohort samples, gene expression data is normalized. Subsequently, a screening process is conducted to identify abnormally expressed genes for each sample (**Fig 1B**). PDRWH consists of three main steps. The first step involves constructing a personalized hypergraph model, where samples are represented as hyperedges, and the mutated genes as well as the abnormally expressed genes in the target sample are described as vertices (**Fig 1C**). In this model, a hyperedge in the hypergraph is capable of connecting multiple vertices (not limited to two nodes as in a simple graph), enabling a more complex representation of relationships between genes and facilitating a comprehensive analysis of association across the cohort

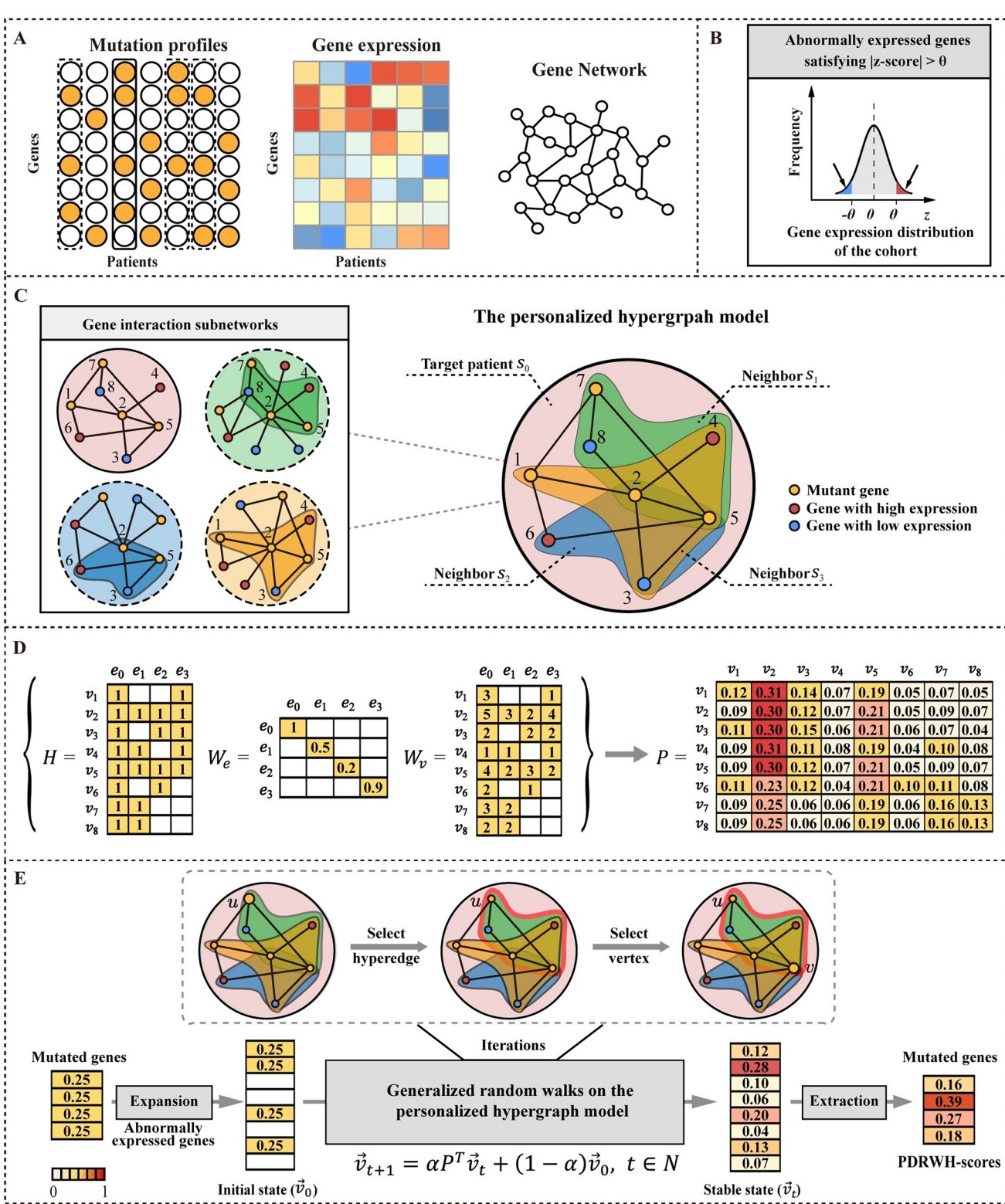

**Fig 1. Overview of PDRWH for prioritizing personalized cancer driver genes.** (**A**) Model input. i) The somatic mutation profiles from the TCGA; ii) The gene expression data of patients; iii) Gene interaction network. (**B**) Pro-processing the gene expression profiles and determining the abnormally expressed genes for each sample. (**C**) Construction of a hypergraph model for each patient. In this model, each hyperedge represents a patient and the vertices incident to each hyperedge represent the mutated genes and abnormally expressed genes in the corresponding patient. (**D**) Computing the transition probability matrix of the random walks on the weighted hypergraph. (**E**) The process of generating PDRWH-scores through random walks on the personalized hypergraph.

samples. As depicted in **S1 Fig,** each hypergraph typically consists of more than 50 hyperedges (samples), indicating that PDRWH generates a relatively dense personalized hypergraph for most individuals. Following this, weights are assigned to hyperedges based on the Pearson correlation of gene expression profiles between the corresponding samples and the target sample. This weighting scheme ensures that patients who closely resemble the target patient have a more substantial impact on the personalized driver gene prediction. The weights of vertices within each hyperedge are set as the degree of the subnetwork induced by genes corresponding to these vertices, characterizing the influence of mutated genes on the abnormally expressed genes in each sample. In the second step, the transition probability matrix is calculated by taking into account the structure of the hypergraph as well as the weights of the hyperedges and vertices. This matrix indicates the probabilities of transitioning from one vertex to another within the hypergraph (**Fig 1D**). The final step entails performing a random walk with restart on the weighted personalized hypergraph. The walker, starting at a present vertex $u$, firstly chooses a hyperedge with probability determined by the weights of hyperedges, and then travels towards any vertex (for example $v$) within the selected hyperedge based on the weights of the vertex in the hyperedge. After the random walks reach a stable state, a stationary distribution of the random walk is achieved. At this point, the PDRWH-scores are generated to quantitatively and rationally prioritize candidate genes (**Fig 1E**). For a target patient, a mutated gene in the target sample should be ranked higher if it is adjacent to many genes that are abnormally expressed in the target sample as well as in a group of other samples from the same cancer type.

## Known drivers have a higher degree of connectivity to abnormally expressed genes in the gene interaction subnetwork

Our approach is built upon a crucial observation: the mutated genes that are adjacent to a higher number of abnormally expressed genes are more likely to have a significant impact. This assumption can be validated through empirical data analysis. To achieve this, we randomly selected some patients from the five cancer types, breast invasive carcinoma (BRCA), kidney renal clear cell carcinoma (KIRC), liver cancer (LIHC), glioblastoma (GBM), and stomach adenocarcinoma (STAD), and created personalized gene interaction subnetworks for each patient by mapping the mutated genes and abnormally expressed genes onto STRINGv10. We then categorized all mutated genes in each patient into two groups based on whether they are known driver genes and analyzed their node degrees distributions in the personalized gene interaction subnetwork using the Satterthwaite approximation t-test (**S2 Fig**). Our analysis revealed that the interactions involved by known driver genes were generally more extensive compared to other mutant genes (p-value < 0.05). Furthermore, when we aggregated samples from all five cancer types into a large cohort, the observed difference became even more statistically significant (p-value $2.2 \times 10^{-16}$).

## PDRWH outperforms existing driver gene prediction methods in identifying known general drivers

We applied PDRWH to the datasets of 16 cancer types from TCGA (**S1 Table**). To evaluate the method, we utilized a union of four well-studied cancer gene databases as a general driver gene reference with a total of 758 genes, including the Cancer Gene Census (CGC) [28], the HiConf cancer gene panels [29], the high-confidence drivers (HCD) [30], and Mut-driver genes defined by the '20/20 rules'. We evaluated the performance of PDRWH's ability to identify known general driver genes based on the top-ranked genes. As shown in the **S2 Table**, in as many as 85% of the total samples, the genes ranked first by PDRWH are known driver

genes. Our approach shows exceptional performance in identifying known driver genes, particularly when considering genes at the top of the predictions, except in SKCM. In LGG and UCEC, this proportion even reaches up to 95.8% and 98.2%. On the whole, approximately 50% of the samples have the gene ranked second as a known driver gene. In SKCM, DNAH8 ranked first in 29% of samples for the personalized prediction. Despite not being listed as a known driver gene, previous studies have indicated its significant association with cancer, thereby categorizing it as potential driver gene [31,32]. As the ranking increases, the proportion of samples where these genes are known driver genes decreases, just as we expected.

Subsequently, we conducted a comparative analysis of PDRWH with four other personalized cancer driver prediction methods (PersonaDrive, Prodigy, DawnRank, and SCS) using five cancer datasets (BRCA, KIRC, LIHC, GBM, and STAD). All the methods utilized the same cancer dataset, which included somatic mutation data and the gene expression data of the tumor samples in TCGA, along with the same gene molecular network STRINGv10 for all the network-based methods. As for the SCS method, we had to limit the analysis to 50 randomly selected patients for each cancer type due to its extended runtime. This allowed us to efficiently run the SCS algorithm while still obtaining meaningful results for comparison with the other methods. At the same time, to highlight the impact of utilizing different data information on algorithm results, two naive methods were introduced as baselines. One method ranks genes based on the mutation frequency among samples, while the other method ranks genes based on the degree in the gene subnetwork induced by mutated genes within the sample. As mentioned above, we utilized the general driver gene reference as the benchmark of known drivers. From **Figs 2** and **S3**, we found that methods utilizing the cohort data (PDRWH, PersonaDrive, and Frequency) consistently outperformed those relying on individual data (Prodigy, DawnRank, and Degree) in terms of average precision, recall, and F1-score. This highlights the significant advantage of leveraging collective data over individual data. Among the three cohort data analysis methods (PDRWH, PersonaDrive, and Frequency), PDRWH exhibits superior performance, achieving an outstanding precision rate of up to 88.4% and 76.0% for genes ranked first in BRCA and KIRC. Although PDRWH's precision for the top-ranked gene in KIRC is slightly lower than PersonaDrive, its superiority becomes more noticeable as the ranking advances, particularly in terms of recall and F1-score. Therefore, PDRWH excels in providing a more precise and rational prioritization of known general driver genes for individual patients.

For each cancer type, we aggregated the personalized candidate driver gene rankings to create a prioritization for the cohort. This allowed us to compare the results of personalized prediction methods with those of cohort-level methods. Using the general reference driver gene list as a benchmark, we generated receiver operating characteristic curves (ROC) and calculated areas under the curve (AUC). In **Figs 3** and **S4**, it is clear that PDRWH outperforms other tools in terms of sensitivity and specificity in identifying known driver genes in BRCA, KIRC, GBM, and STAD. Additionally, the AUC values of PDRWH are consistently at the highest level for BRCA, LIHC, and GBM, while ranked second compared to other tools in KIRC and STAD.

## PDRWH achieves reliable results in identifying known tumor-specific drivers

Considering the diversity among cancer types, tumor-specific drivers hold more concern than the general drivers across different tumor types. Therefore, it is highly valuable to assess the ability of methods to accurately identify cancer-specific driver genes. To achieve this, we downloaded a set of tumor-specific driver genes from the IntOGen database as benchmarks. To further support the efficiency of PDRWH by statistical significance, the enrichment *p*-

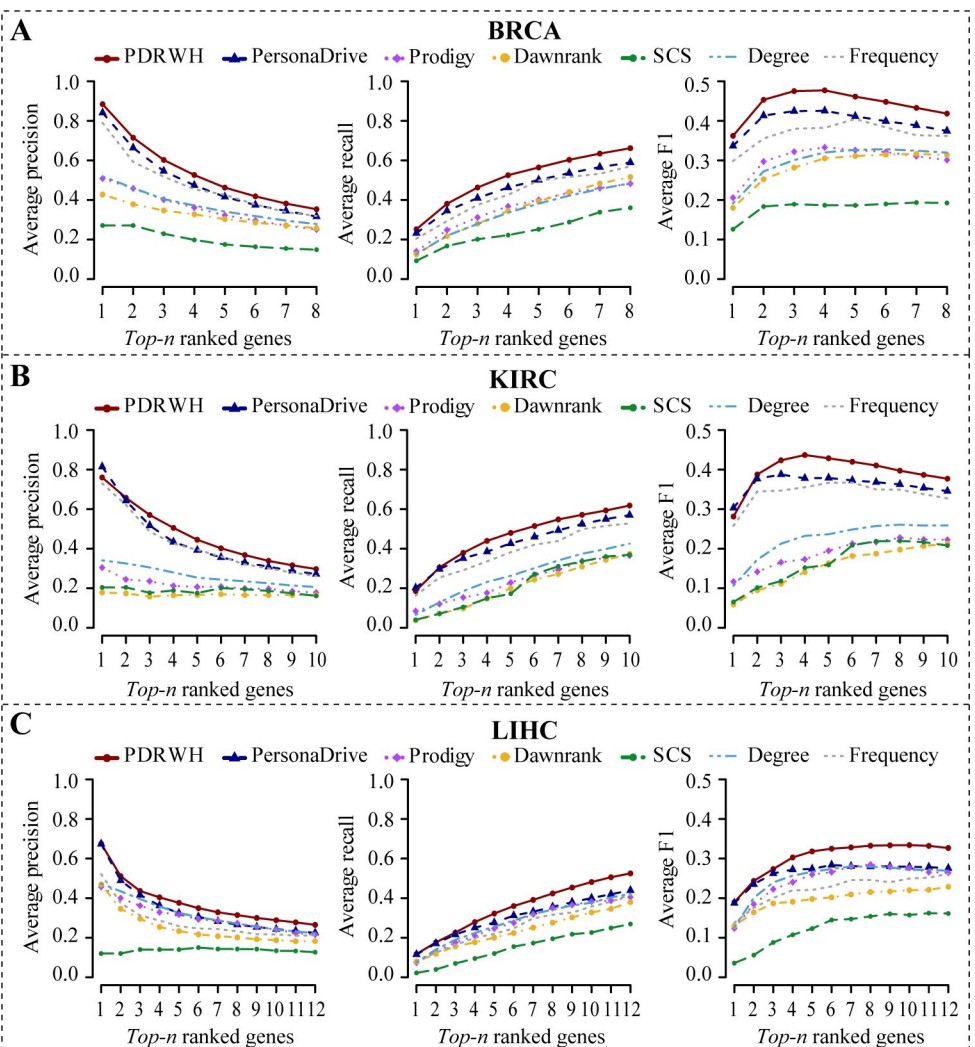

**Fig 2. Comparison of the PDRWH with the other personalized prediction methods.** The average precision, recall, and F1-score for (**A**) the BRCA dataset, (**B**) the KIRC dataset, and (**C**) the LIHC dataset, are plotted as a function of the number of *top-n* ranked genes involved in the calculation of the scores. The general driver gene list is used as the reference set.

values of predicted genes in the tumor-specific driver gene lists were evaluated using the hypergeometric test, representing the significance of tumor-specific driver genes rediscovered by PDRWH. The computational details are provided in the Methods and Materials section. From the result of **S5 Fig**, we can observe that PDRWH shows some variability in performance across 16 cancer types. With the exceptions of LUSC and SKCM, as many as three-quarters of the samples from each cancer type demonstrate enrichment in the tumor-specific driver gene list.

For comparison, we used a naive method that randomly selected mutated genes as predicted personalized drivers as a baseline. From **Fig 4A**, we can find that the five personalized driver prediction methods show higher percentages of significant samples for identifying the cancer-specific driver genes than randomly chosen. In comparison to other methods, PDRWH consistently outperforms. For instance, in the case of BRCA, our method achieves significant enrichment in BRCA-specific cancer drivers for 85.19% of the samples. Similarly,

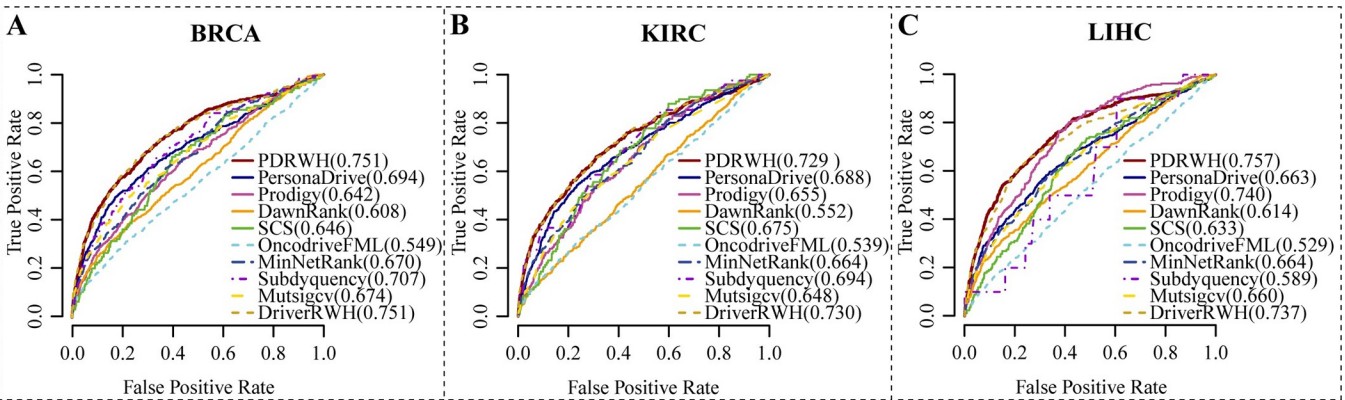

**Fig 3. Prediction performance of five personalized prediction methods as well as five cohort prediction methods.** (**A–C**) ROC plots of results on the different cancer types based on the general reference driver set. The solid lines represent the personalized prediction methods (PDRWH, PersonaDrive, Prodigy, DawnRank, and SCS). The dashed lines indicate the cohort-level prediction methods (OncodriveFML, MinNetRank, MutsigCV, Subdyquency and DriverRWH). The numbers in parentheses behind the methods are the AUC values of the corresponding method.

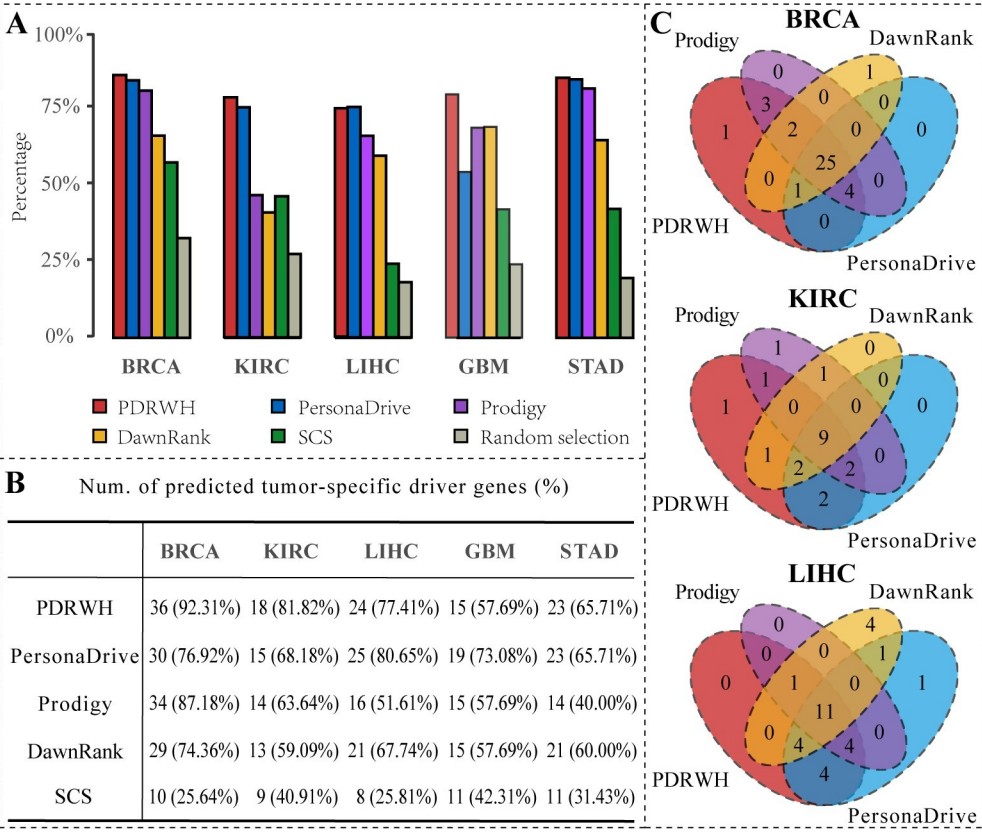

| | BRCA | KIRC | LIHC | GBM | STAD |
|---|---|---|---|---|---|
| PDRWH | 36 (92.31%) | 18 (81.82%) | 24 (77.41%) | 15 (57.69%) | 23 (65.71%) |
| PersonaDrive | 30 (76.92%) | 15 (68.18%) | 25 (80.65%) | 19 (73.08%) | 23 (65.71%) |
| Prodigy | 34 (87.18%) | 14 (63.64%) | 16 (51.61%) | 15 (57.69%) | 14 (40.00%) |
| DawnRank | 29 (74.36%) | 13 (59.09%) | 21 (67.74%) | 15 (57.69%) | 21 (60.00%) |
| SCS | 10 (25.64%) | 9 (40.91%) | 8 (25.81%) | 11 (42.31%) | 11 (31.43%) |

**Fig 4. The performance of PDRWH and other four methods for identifying the known tumor-specific driver genes.** (**A**) The percentage of patients whose predicted personalized drivers are significantly enriched in the known tumor-specific driver gene list. (**B**) Comparison of the number of predicted tumor-specific driver genes by various methods and the recall ratio. (**C**) Overlap among the tumor-specific cancer drivers predicted by different methods for BRCA, KIRC, and LIHC.

for KIRC, LIHC, GBM, and STAD, the percentages are 77.41%, 74.16%, 78.67%, and 84.86% respectively. These results indicate that a considerable number of samples support the reliable outcomes produced by PDRWH, highlighting its effectiveness in predicting tumor-specific driver genes for individuals.

Assuming that the data of a particular cancer type from TCGA is sufficiently representative of the majority of samples for that cancer, the known tumor-specific driver genes should be detected in the corresponding cancer samples. For each cancer type, we collected the predicted personalized driver genes of all tumor samples and compared PDRWH with four other methods based on their overlap with the known tumor-specific cancer drivers. The number of known tumor-specific driver genes identified by different methods is illustrated in **Fig 4B** for each cancer type. PDRWH successfully detected 36 out of 39 known BRCA-specific driver genes and 18 out of 22 KIRC-specific drivers, demonstrating its superiority over the other methods. In the remaining three cancer types, our method exhibited similar performance to PersonaDrive while still outperforming DawnRank, Prodigy, and SCS.

Furthermore, we analyzed the overlap and difference of the identified known tumor-specific driver genes between different methods (**Figs 4C and S6A**). Since SCS utilized insufficient samples, its results were not included in the comparison. Our findings revealed that PDRWH was able to detect a majority of the known drivers predicted by other methods. It is noteworthy that PDRWH also identified known tumor-specific drivers that were missed by other methods, such as *ABL2* for BRCA, *SETBP1* for KIRC, *NIN* and *TOP2A* for STAD (**S3 Table**). This suggests that PDRWH can serve as a complementary approach to other methods to promote tumor-specific driver gene identification.

## PDRWH efficiently identifies both common and rare drivers

One of the advantages of our method is its ability to identify both common and rare driver genes, demonstrating a balanced performance that effectively trades off algorithm generalizability and specificity. To further demonstrate this ability, we divided the top-ranked predicted driver genes into two categories based on the frequency of occurrence in the respective cancer-type cohort: common and rare. Genes with mutation frequency $\geq$2% are classified as common drivers, while those with a mutation frequency <2% are classified as the rare. The results are summarized in **Figs 5A–5C and S6B.** The majority of genes ranked at first are known high-frequency driver genes. As the ranking increases, even if the proportion of known high-frequency drivers decreases, the known low-frequency drivers always have a relatively stable proportion, which indicates the PDRWH's capability in detecting known rare drivers. **Figs 5D–5F and S6C** show the scatter plots of the gene mutation frequency versus the frequency that appears as the personalized driver genes in patients. The mutation frequencies of the known driver genes (red dots) exhibit a remarkably high concordance with those drive genes predicted by PDRWH (Pearson R-square > 0.99, p-value < 2.2e-16). Across the five cancer types, the potential driver gene *TTN* with high mutation frequency is the most promising, which has been observed to be involved in several cancer functions and to become an effective predictor for overall survival and chemotherapy response [31]. In LIHC, *ALB*, a novel potential driver gene predicted by PDRWH, has been proposed as an effective biomarker for cancer detection [32]. In addition, all the predicted personalized driver genes but not the known drivers are also enriched in multiple cancer-related pathways based on the Database for Annotation, Visualization and Integrated Discovery (DAVID) online database [33] and Kyoto Encyclopedia of Genes and Genomes (KEGG) database [34]. The results of the enrichment analysis are shown in **S4 Table** and **S7 Fig**.

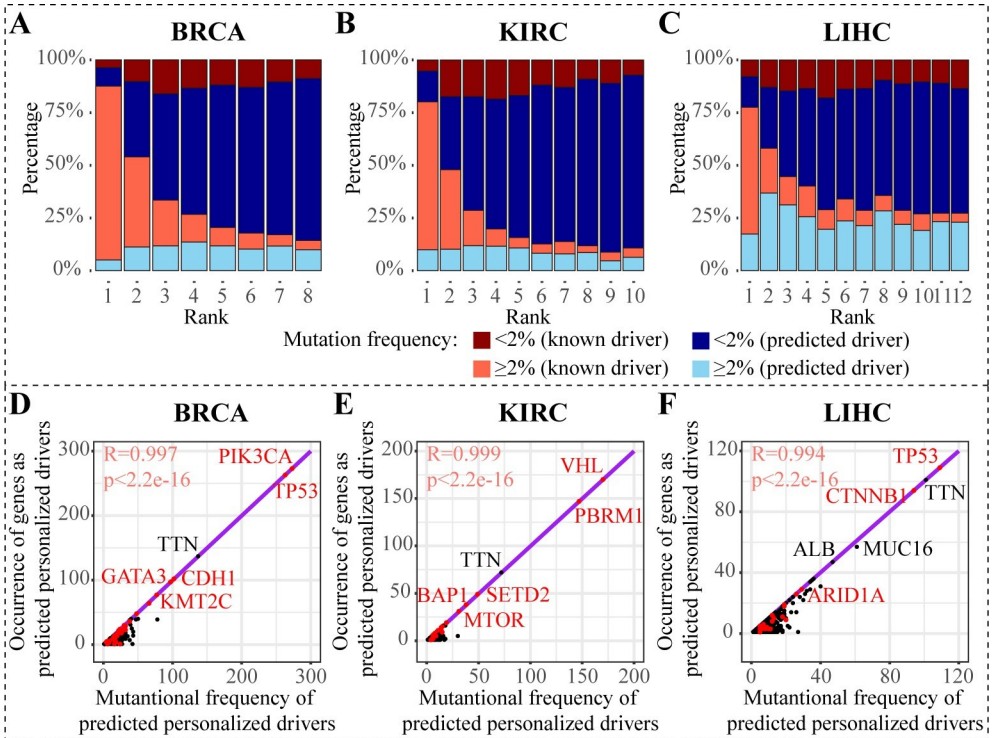

**Fig 5. PDRWH identifies both common and rare drivers.** (**A-C**) Distribution of mutation frequency of top genes predicted by PDRWH. The i-th column in the plot represents the distribution of mutation frequency of the genes which ranked at the i-th in the predicted drivers. Each range of mutation frequency is further classified into whether the genes are known drivers in the reference set. (**D-F**) Scatter plots about mutation frequency of potential drivers and the occurrence of genes as predicted driver gene. Known tumor-specific driver genes are represented as red dots and others are represented as black dots. Purple lines constructed by known tumor-specific driver genes are the regression lines.

## Subtypes recovered by expressions of predicted rare drivers are significantly associated with patient survival

Considering that the state of driver genes is supposed to reflect their phenotypic impact on carcinogenesis, we further verified the ability of the personalized drivers identified by PDRWH in stratifying tumor samples. Employing unsupervised K-means clustering, we separated the tumor samples into different subtypes based on the gene expression of the predicted personalized drivers. The number of clusters for each cancer type was determined using a CDF (Cumulative Distribution Function) curve. We suggested that there is a distinction in the effectiveness of the rare (<2%) drivers and the common (≥2%) drivers used for stratifying patients. From **Figs 6A, 6B and S8**, the expression of known tumor-specific drivers was unable to identify subtypes correlated with patient survival regardless of which part of the gene set was used. When the expression profiles on predicted drivers with high frequency were used, we could obtain a significant survival analysis result among KIRC patients only (**Fig 6C**). Notably, predicted driver genes with low frequency have significant prognostic values for survival in BRCA, KIRC, and LIHC (**Fig 6D**).

Similar results were also obtained in GBM and STAD (**S9 Fig**). It should be noted that the numbers of subtypes (we recognized five subtypes for BRCA, four subtypes for KIRC, four subtypes for LIHC, and three subtypes for both GBM and STAD) are consistent with the approbatory numbers of subtypes in literature [35–39]. To sum up, the rare drivers predicted

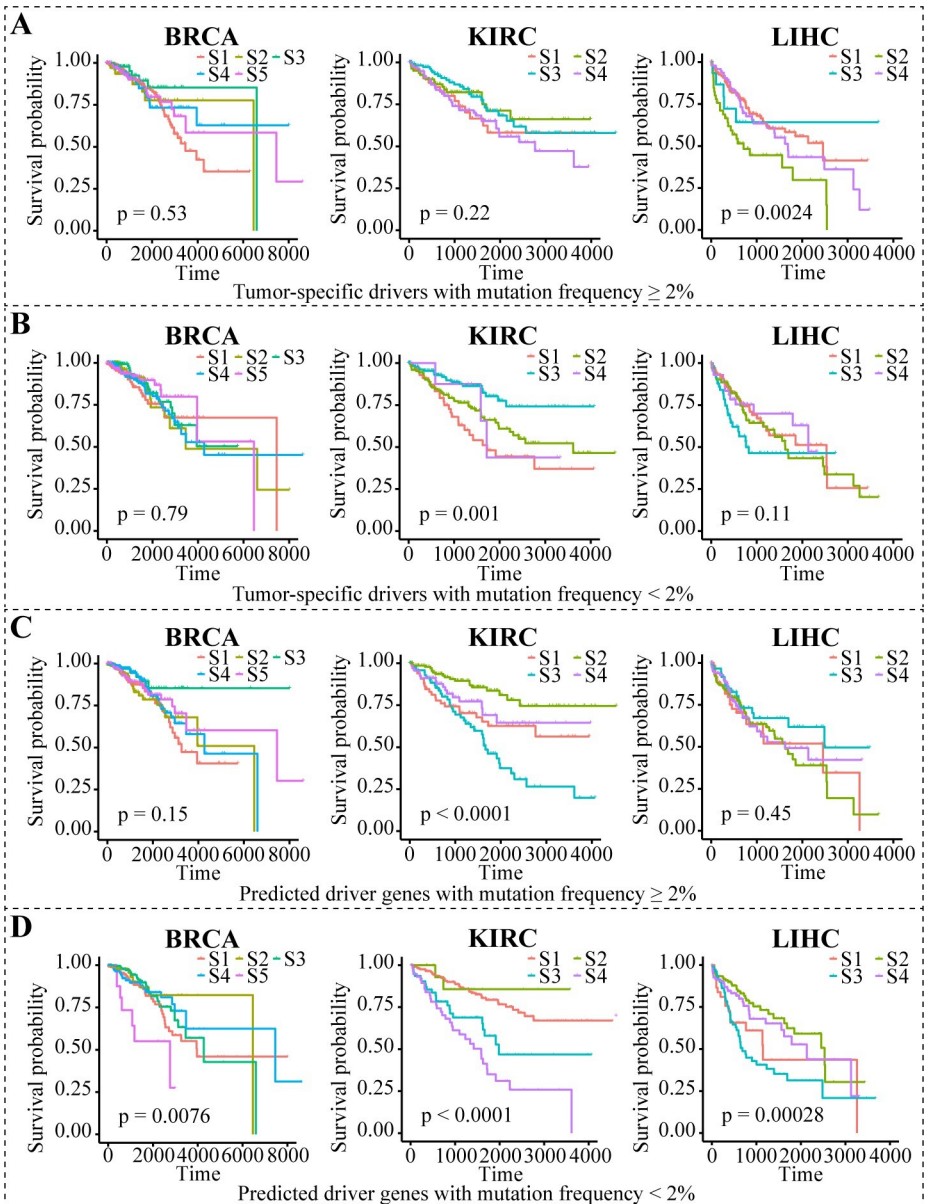

**Fig 6. The survival curves for subtyping BRCA, KIRC, and LIHC using the gene expression data.** (**A**) In different cancer types, the expression data of known tumor-specific drivers with mutation frequency ≥ 2% were used in subtyping patients. Different subtypes (S1, S2,...) are indicated by different colored lines. (**B-D**) The similar analysis based on expression data of genes that are known tumor-specific drivers with mutation frequency < 2%, predicted driver genes with mutation frequency ≥ 2%, and predicted driver genes with mutation frequency < 2% respectively.

by PDRWH are more conducive to subtype clustering, and we can assume that our methods capture more precise causative events for carcinogenesis in individuals.

## Clinical application of the predicted driver genes

We next evaluated whether our predicted personalized driver genes can provide useful information to the oncologist in deciding on therapy. For each patient, the number of predicted personalized drivers in Therapeutically Applicable Research to Generate Effective Treatments

(TARGET, 135 actionable genes), the Drug-Gene Interaction database (DGIdb, 1387 drug-gable genes) [40], and the union of the two sets (1407 actionable or druggable genes) was counted. As shown in **Figs 7** and **S10**, more than three-quarters of patients have at least one actionable gene, and the majority of patients contain more than one druggable driver. As a case, in the union set, there are only 3.58% of patients on BRCA, 2.48% on KIRC, and 1.12% on LIHC without any actionable genes or druggable genes. These results confirm that the results predicted by PDRWH may be reasonable and useful references in individually tailored therapy.

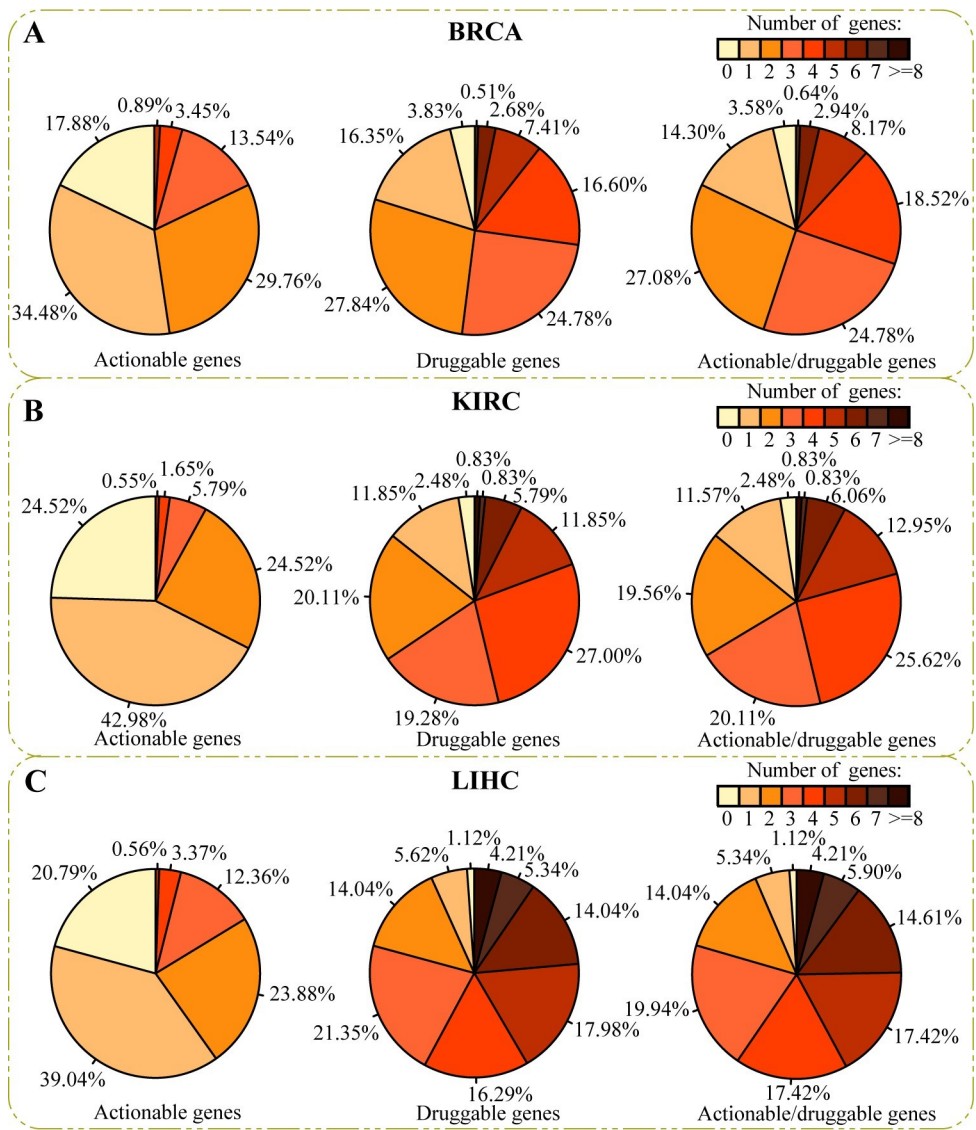

**Fig 7. Distribution of the number of predicted personalized driver genes in TARGET and DGIdb.** (**A**) For cancer type BRCA, the first pie chart shows the distribution of the number of predicted personalized driver genes in TARGET. Restricted to predicted personalized drivers predicted by PDRWH, there are 17.88% of patients with not less than three actionable driver genes. The second pie chart shows the distribution of the number of predicted personalized driver genes in DGIdb. There are more than 50% of patients with not less than three druggable personalized drivers. The third pie chart is the distribution of the number of predicted personalized driver genes in the union of the two sets. (**B-C**) The similar pie charts display for cancer type KIRC and LIHC.

### Experimental validation of predicted novel cancer drivers

Finally, we performed in vitro cell-based assays for a novel drive gene identified by PDRWH (**S1 Text**). By way of illustration, Low-density lipoprotein receptor-related protein 1 (LRP1) was predicted as a driver gene in Human gastric cancer cells (GC) by our method and PersonaDrive. This gene was not detected by widely used methods, such as DawnRank, Prodigy, and SCS, and was not presented in the known driver gene list. As shown in **Fig 8A and 8B**, *LRP1* was up-regulated in GC cells and GC tissues, especially, higher expressed in the HGC-27, MGC-803, and AGS cells. *LRP1* was also negatively associated with the overall survival rates of patients with GC (**Fig 8C**). The overall survival analysis of *LRP1* was based on Gene Expression Profiling Interactive Analysis (GEPIA) [41]. To investigate the potential cancer-related roles of *LRP1*, loss-of-function assays were performed in HGC-27 cells. After the transfection of three siRNAs respectively, which could produce specific weak knock (knockdown) effects on the *LRP1* gene, *LRP1* expression in the three experimental groups was significantly on the decline at the mRNA and protein levels (**Fig 8D**). By Wound healing assay, we also observed that the knockdown of *LRP1* suppresses the metastatic ability of HGC-27 cells (**Fig 8E**). Besides, the knockdown of *LRP1* increased the apoptosis rate of gastric cancer cells (**Fig 8F**), and inhibited cell proliferation reducing the proliferation rate of HGC-27 cells from 4.4% in controls to 1.2%~1.8% in experimental groups (**Fig 8G**). Furthermore, depletion of *LRP1* could induce G1 and G2 phase arrest (**Fig 8H**). These collective preliminary results indicate that *LRP1* predicted by PDRWH as a personalized driver gene is potentially involved in the development of GC.

## Discussion

Identifying personalized driver genes that lead to particular cancer initiation and progression of individual patients is a crucial part of precision medicine. In this study, we have presented an unsupervised learning method to identify patient-specific driver genes by leveraging genome and transcriptome datasets from a cohort. PDRWH, similar to many unsupervised algorithms, can directly uncover hidden patterns and structures in data without the need for explicit model training or a large number of labeled genes, aiming to provide comprehensive support for the analysis of driver genes. The novelty of this study lies in the introduction of the concept and methodology of hypergraph random walks to predict personalized driver genes. The hypergraph model offers substantial benefits in terms of data integration and interpretability. By grouping patient-specific mutated genes and abnormally expressed genes within the corresponding hyperedge, the hypergraph model allows for a comprehensive representation of association among genes across multiple samples simultaneously, rather than separately. The random walk algorithm on the hypergraph is tailored to generate a quantitative assessment of the influence on the gene interaction network within the target sample and its neighboring samples, facilitating the systematic prioritization of candidate personalized driver genes. Comparisons based on the TCGA have demonstrated the superior performance of PDRWH over other computational methods in identifying the known cancer drivers. We believe our method will complement existing driver identification methods and will help us discover potential personalized drivers, especially those rare drivers that often escape detection by other methods.

One limitation of the current model is that it relies on a broad-context molecular network rather than a patient-specific one. As a result, it overlooks regulation information that is specific to individual patients, which could potentially lead to false positives in the results. Moreover, PDRWH focuses on prioritizing single genes, while it is well-established that genes often collaborate to drive cancer initiation and progression. Therefore, there is a clear need for computation methods to elucidate how these genetic aberrations collaborate to induce

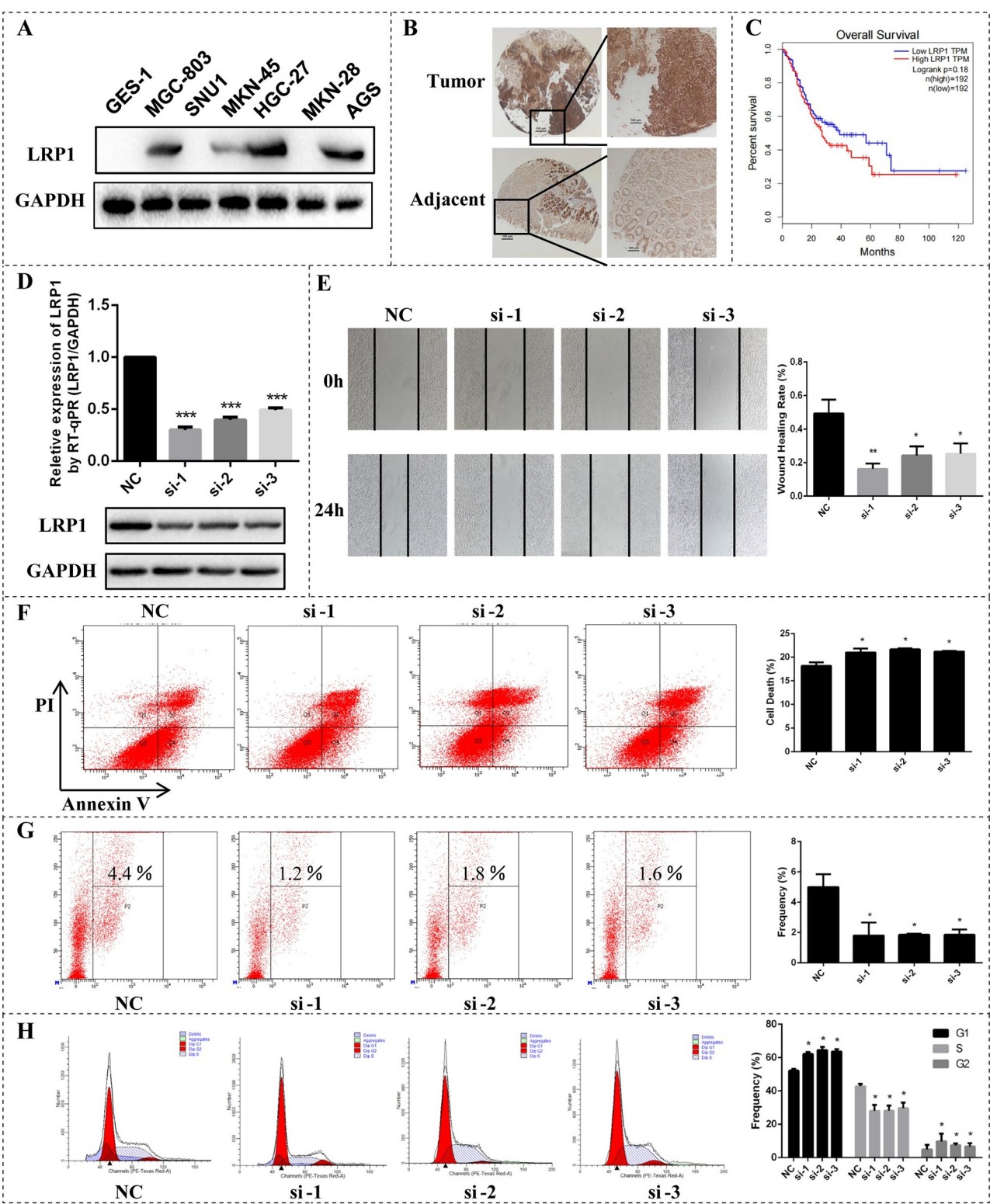

**Fig 8. In vitro assays of a novel driver gene *LRP1* predicted by PDRWH.** (**A**) The expression of *LRP1* was detected between GES-1 and GC cells. (**B**) The expression of *LRP1* was detected between GC tissues and adjacent tissues using Immunohistochemical analysis. (**C**) Overall Survival analysis of *LRP1* based on GEPIA. (**D**) HGC-27 cells transfected with siRNA by real-time PCR and Western Blot. (**E**) Wound healing assay following knockdown of *LRP1* in HGC-27 cells. (**F**) Apoptosis detection for HGC-27 cells transfected with siRNA. (**G**) Proliferation detection for HGC-27 cells transfected with siRNA using EdU assay. (**H**) Cell cycle profile of control and *LRP1* knockdown cells. *GAPDH* protein is used as control. All cell assays were performed in triplicate. The error bars indicate SD of three independent experiments. *P < 0.05, **P < 0.01 using the two-sided Student's t test.

transcriptional abnormalities, and ultimately lead to the onset of cancer. Additionally, our current model primarily focuses on point mutations, including single-nucleotide variants (SNVs) and short insertions or deletions (indels), due to their prevalence, ease of detection, and potential as genetic markers for specific phenotypes that promote tumor formation. However, the impact of other somatic alterations like amplifications, genomic rearrangements, and epigenetic silencing, which are also crucial in tumorigenesis, has not been considered. Integrating information on all these alterations would improve the identification of driver genes in cancer.

Despite these limitations, PDRWH has demonstrated reliable performance in inferring personalized driver genes, which is promising for discovering potential causal genetic variants that would be obscured by tumor heterogeneity. In the future, we expect that PDRWH will assist in the development of optimal personalized treatment.

## Methods and materials

### Data resources

In this work, we use two types of genomic data from a cohort: somatic mutation data, which includes non-synonymous point mutations and insertions/deletions (indels) in coding regions, and gene expression profiles. We downloaded 16 cancer datasets that contained a sufficient number of samples with both mutation and gene expression data (>150 samples) from the TCGA data portal [42] through the Xena platform [43]. The samples with less than three mutated genes in the cohort were filtered out. PDRWH also uses a gene interaction network: a global PPI network taken from STRINGv10 [44]. This network includes 17084 genes and 3513941 interactions. Information about the databases is given in the **S1 Table**. It should be noted that we primarily used five common types of cancer as examples to demonstrate the evaluation of the algorithm's performance, including breast invasive carcinoma (BRCA), kidney renal clear cell carcinoma (KIRC), liver cancer (LIHC), glioblastoma (GBM), and stomach adenocarcinoma (STAD). Additionally, we also provided a summary of the performance of PDRWH on the other cancer types.

### Pre-processing

In a given cohort gene expression profile $X = \{x_{i,j}\}$, where the rows represent genes and columns represent patient samples, the gene expression values are processed using z-score normalization by the following formula:

$$x'_{i,j} = \frac{x_{i,j} - \mu_i}{\sigma_i} \tag{1}$$

where $\mu_i$ and $\sigma_i$ are the mean expression values and standard deviation of the gene $i$. To identify the abnormally expressed genes for each patient, a threshold $\theta$ is set for each gene, which is the smaller value between the absolute values of the 5% and 95% quantiles among $\{x'_{i,j}\}$, to pick the significant high or low expression values of the standard gene expression profile $X'$. A gene $i$ is regarded as an abnormally expressed gene of sample $j$ if its absolute value of $x'_{i,j}$ is more than $\theta$, which indicates that the gene is expressed quite differently in this sample compared to the other samples. The advantage of this definition of abnormally expressed genes is that it does not rely on the existence of paired normal and tumor data of the same patient or background gene expression profile from healthy samples. It provides a robust method to detect genes with distinct expression patterns within the cohort. For each sample, the number of genes contained in the set of abnormally expressed genes is approximately about 400 to 1600.

## Construction of personalized hypergraph model

At first, a personalized weighted hypergraph model is constructed to accurately capture the implicit inherent similarity of samples and the association between the mutated genes and abnormally expressed genes. For a target sample $s_0$, if a sample shares at least one co-mutant gene with this one, it is defined as a neighbor of that patient. Let $S = \{s_1, \ldots, s_n\}$ denote the set of neighbor samples of the target sample. Then, we defined the patient-specific hypergraph $H(V,E)$, where $V$ is a set of vertices representing all mutant genes and abnormally expressed genes of $s_0$, and $E = e_0 \cup \{e_1, \ldots, e_n\}$ is a set of hyperedges. Hyperedge $e_0$ represents the target sample $s_0$, which is incident with node $v$ if this gene is mutated or abnormally expressed in this sample. Likewise, hyperedges $e_1, \ldots, e_n$, corresponding samples $s_1, \ldots, s_n$, are incident with their mutant genes and abnormally expressed within vertex set $V$. The incidence matrix $H \in \mathcal{R}^{|V| \times |E|}$ is defined as follows:

$$h(v, e) = \begin{cases} 1, & \text{if } v \in e \\ 0, & \text{otherwise} \end{cases} \tag{2}$$

which indicates whether vertices are incident with the hyperedges. According to the assumption that patients with similar gene expression profiles may have the more similar pathogenic mechanism to each other than the rest of the patients, the most closely related patient in terms of its gene expression profile will contribute much more to the prediction of driver genes compared to other patients. Therefore, the weight of a hyperedge should be an increased function with their correlation to the target sample. A fairly standard choice for the weights is:

$$w_e = exp\left(-\frac{\|1 - \rho(e, e_o)\|^2}{2\delta^2}\right), e \in E \tag{3}$$

in which $\rho(e, e_o)$ is the Pearson correlation of gene expression profiles between sample $s$ and the target sample $s_0$, and $\delta$ is the bandwidth parameter (default $\delta = 0.1$) controlling how quickly the weight of a sample falls off with the distance of $s$ from the query point $s_0$. Here, if $\rho(e, e_o)$ is close to 1, $w_e$ will also be close to 1, implying that this sample has a high impact in the evaluation of driver genes of target sample $s$. On the contrary, if $\rho(e, e_o)$ is small (e.g., close to 0), $w_e$ will be relatively small too. In this case, the corresponding neighbor will have a weak contribution to the determination of the driver genes of the target sample. Then the weight matrix of hyperedge is defined as the diagonal matrix:

$$W_e = diag\{w_e | e \in E\} \tag{4}$$

To model the relationship between the mutated genes and the abnormally expressed genes, we project the mutant genes and abnormally expressed genes in each hyperedge onto a human gene interaction network. PDRWH views the gene network as an undirected graph. For gene $v_i$ and $v_j$ in hyperedge $e$, an edge exists if the two genes interact in the gene interaction network. This way, the vertices in each hyperedge $e$ induce a corresponding gene interaction subnetwork $N_e$. Since the driver genes tend to adjacent more abnormally expressed genes in the subnetwork $N_e$ (**S1 Fig**), the weight of node $v$ in the hyperedge $e$, denoted as $w(v_e)$, can be set as its degree in $N_e$. The matrix $W_v \in \mathcal{R}^{|V| \times |E|}$ is defined as follows:

$$w(v, e) = \begin{cases} w(v_e), & \text{if } v \in e \\ 0, & \text{if } v \notin e \end{cases} \tag{5}$$

Since the interaction of isolated vertices with other genes in the network is unknown, the weights of the isolated vertices are set to a small value of 0.01 instead of 0. Then, the degrees of

vertices and hyperedges in the weighted hypergraph are defined as:

$$d(v) = \sum_{e \in E} h(v, e) w_e, v \in V \tag{6}$$

$$\delta(e) = \sum_{v \in V} w(v, e), e \in E \tag{7}$$

## Transition probability matrix of the random walks on hypergraph

A random walk on a hypergraph $H(V, E)$ is similar to the classic random walk, where transitions occur between two incident vertices in the hyperedge [45]. Specifically, the movement between vertices is modeled as a discrete-time Markov chain based on predefined transition probabilities. A standard formulation for a hypergraph random walk can be broken down into two steps. Given the current state $\overrightarrow{v}_t$:

i.  Starting at a vertex $u$, a hyperedge is selected with a probability determined by the weights of hyperedges $w_e$.

ii. a vertex $v$ is chosen from the selected hyperedge $e$. The walker can travel to any nodes within the selected hyperedge based on the weights of the vertex in the hyperedge $w(v, e)$.

Thus, transition probabilities from vertex $u$ to vertex $v$ are calculated as follows:

$$p(u, v) = \sum_{e \in E} \frac{h(u, e) w_e}{\sum_{\hat{e} \in E} h(u, \hat{e}) w_{\hat{e}}} \frac{w(v, e)}{\sum_{\hat{v} \in V} w(\hat{v}, e)} \tag{8}$$

which can be written in an alternative matrix form:

$$P = D_v^{-1} H W_e D_{ve}^{-1} W_v^{\ T} \tag{9}$$

where $H$, $W_e$, and $W_v$ are defined as previously mentioned, while $D_v$ and $D_{ve}$ represent the diagonal matrix for the degrees of vertices and hyperedges, with $d(v)$ and $\delta(e)$ being the respective diagonal elements.

## Generalized random walks on the personalized hypergraph model

We implement a random walk with restart on the personalized hypergraph by adding a damping factor. Specifically, all the mutated genes in the target sample $s_0$ are assumed to have an equal probability of being driver genes. Therefore, an equal probability of $\frac{1}{n}$ is assigned to each of the mutated genes in $s_0$ initially, where $n$ represents the number of mutated genes in $s_0$. The initial values of the abnormally expressed genes are set to zero. Let the initial column vector be denoted as $\overrightarrow{v}_0 \in \mathcal{R}^{|V|}$. The process can be mathematically represented by the following formula:

$$\overrightarrow{v}_{t+1} = \alpha P^T \overrightarrow{v}_t + (1 - \alpha) \overrightarrow{v}_0, t \in N \tag{10}$$

where the $i$-th element of state $\overrightarrow{v}_t$ represents the probability that the walker moves to node $i$ at step $t$. The damping factor, $\alpha$ ($0 < \alpha < 1$), is introduced to ensure the graph satisfies ergodic conditions [45]. In our study, we have empirically set the damping factor $\alpha$ to 0.85. The term $\alpha P^T \overrightarrow{v}_t$ in the formula means that the random surfer may transition to one of the adjacent vertices, while $(1 - \alpha) \overrightarrow{v}_0$ represents a vector introducing the probability of teleporting the random walk back to the initial state. After several iterations of the random walk, the distribution vector $\overrightarrow{v}$ stabilizes when the difference between $\overrightarrow{v}_{t+1}$ and $\overrightarrow{v}_t$ measured by L1 norm falls below a small $\varepsilon$ (default $10^{-6}$). PDRWH algorithm generally converges within ten times iterations. The stationary probability indicates the likelihood of genes being personalized cancer drivers of the target sample.

Finally, the values of mutated genes in the stationary probability vector are preserved and then normalized to generate the PDRWH-score. PDRWH-scores can be ranked in descending order to prioritize the personalized candidate driver genes.

## Comparison to other methods

We utilized two benchmarking measures for comparison of the methods' ability to identify known personalized driver genes. One benchmark is the ability to recapitulate many of the well-studied general cancer-associated genes. We assembled a general driver list of 758 known cancer driver genes from various sources, including the Cancer Gene Census (CGC) [28], the HiConf cancer gene panels [29], the high-confidence drivers (HCD) identified by a rule-based method [30], and Mut-driver genes defined by the '20/20 rules', which identifies driver genes based on the characteristic mutational patterns for oncogenes and tumor suppressor genes [46]. This list served as an approximate benchmark of known general drivers for validation. Then, we defined personalized drivers predicted by PDRWH as the *top-n* ranked genes, where *n* was assigned as twice the median of number of mutated genes in the general driver set across the population of patients [21]: 8 for breast cancer (BRCA), 10 for kidney clear cell carcinoma (KIRC), 12 for liver cancer (LIHC), 8 for glioblastoma (GBM), and 16 for stomach cancer (STAD). We used the modified REA strategy proposed by PersonaDrive [21] for a comparison of PDRWH with six personalized prediction methods (DawnRank, Prodigy, SCS, PersonaDrive, Degree and Frequency) across five cancer types from TCGA. For each sample, the identified cancer drivers in the general driver list were adopted to compute the Precision, Recall, and F1-score. Three measurements were generated for each individual:

$$\text{Precision} = \frac{|\text{genes in reference list} \cap \text{genes predicted by computational methods}|}{|\text{genes predicted by computational methods}|} \quad (11)$$

$$\text{Recall} = \frac{|\text{genes in reference list} \cap \text{genes predicted by computational methods}|}{|\text{genes in reference list}|} \quad (12)$$

$$\text{F1}_{\text{score}} = 2 * \frac{\text{Precison} * \text{Recall}}{\text{Precison} + \text{Recall}} \quad (13)$$

in which $|\cdot|$ is the number of genes in a set. The averaged values were calculated for the sake of comparison. To predict driver genes for a cohort and compare them with other cohort-level methods, we utilized an adapted version of PageRank, considering the personalized driver gene ranking score as the voters' preference for candidate driver genes. Using the aforementioned general driver genes as a benchmark, we generated receiver operating characteristic (ROC) curves and calculated areas under the curve (AUCs) to evaluate the true positive and false positive rates. All the details of this study are provided in the **S1 Text**.

The second benchmark involves the identification of tumor-specific driver genes. As there is a remarkable discrepancy among different cancer types, we downloaded a list of tumor-specific driver genes from the IntOGen database [47]. This list is considered to be the best trade-off between sensitivity and specificity among those currently available (**S5 Table**). Given the *top-n* predicted personalized drivers, an enrichment analysis of the personalized driver genes was performed using the hypergeometric test:

$$P(X = m) = \frac{\binom{M}{m}\binom{N-M}{n-m}}{\binom{N}{n}} \quad (14)$$

where *N* represents the total number of genes in a patient, *M* is the number of genes in the

known tumor-specific driver gene list, $n$ is the number of predicted personalized driver genes of the patient, and $m$ is the number of overlapping genes between the known tumor-specific driver genes and predicted personalized driver genes of the individual. If $P(X{\geq}m){<}0.05$, it indicates that the predicted driver genes for this patient is significantly enriched in known driver genes. Additionally, we investigated the consistency and differences in the identified tumor-specific driver genes among different methods.

## Supporting information

**S1 Text. Supplementary material for "A novel hypergraph model for identifying and prioritizing personalized drivers in cancer".**
(DOCX)

**S1 Fig. The distribution of neighbor numbers involved in modeling a personalized hypergraph.** The n in parentheses represents the number of tumor samples.
(TIF)

**S2 Fig. The degree of known driver with the other genes in gene interaction subnetwork.** (**A-E**) Comparison in randomly selected tumor patients. Each subnetwork is induced from STRINGv10 PPI network by the mutant genes and abnormally expressed genes of that patient. (**F**) Cumulating the result of all the patients in a large cohort consisting of 2022 tumor samples across five cancer types. * P < 0.05, ** P < 0.01 *** P < 0.001 and **** P < 0.0001using the Satterthwaite approximation t test.
(TIF)

**S3 Fig. Comparison of the PDRWH with other personalized prediction methods.** The average precision, recall, and F1-score for (**A**) the GBM dataset and (**B**) the STAD dataset, are plotted as a function of the number of *top-n* ranked genes involved in the calculation of the scores. The general driver gene list is used as the reference set.
(TIF)

**S4 Fig. Prediction performance of five personalized prediction methods as well as four cohort prediction methods.** (**A-B**) ROC plots of results on the five cancer types based on the general reference driver set. The solid lines represent the personalized prediction methods (PDRWH, DawnRank, SCS, PRODIGY and PersonaDrive). The dashed lines indicate the cohort-level prediction methods (OncodriveFML, MinNetRank, Subdyquency, MutsigCV and DriverRWH). The numbers in parentheses behind the methods are corresponding AUC values.
(TIF)

**S5 Fig. The p-values of personalized drive genes enriched in tumor-specific drive genes on 16 cancer datasets.**
(TIF)

**S6 Fig. The known driver genes and potential driver genes predicted by PDRWH.** (**A**) Overlap among the tumor-specific cancer drivers predicted by different methods in GBM and STAD. (**B**) Distribution of mutation frequency of top genes predicted by PDRWH. The i-th column in the plot represents the distribution of mutation frequency of the genes which ranked at the i-th in the predicted personalized drivers. Each range of mutation frequency is further classified into whether the genes are known drivers in the general reference driver gene list. (**C**) Scatter plots about mutation frequency of potential drivers and the occurrence of genes as predicted driver gene. Known tumor-specific driver genes are represented as red dots and others are represented as black dots. Purple lines constructed by known tumor-specific

driver genes are the regression lines.
(TIF)

**S7 Fig. Enrichment analysis of potential driver genes in KEGG pathways.** The vertical axis represents the id of KEGG pathway, such as "hsa04020: Calcium signaling pathway" and "hsa05022: Pathways of neurodegeneration-multiple diseases". The ids and names of KEGG pathways can be found in **S4 Table**. And "GeneRatio" represents the ratio of the number of genes enriched in the target pathway to the gene list. (**A-E**) The potential driver genes predicted in the cohort by PDRWH enriched in part of KEGG pathways.
(TIF)

**S8 Fig. Tumor stratification using the gene expression of known tumor specific drivers.** (**A-E**) Tumor stratification using the gene expression of known tumor specific drivers in BRCA, KIRC, LIHC, GBM and STAD respectively. Different subtypes (S1, S2,...) are indicated by different colored lines.
(TIF)

**S9 Fig. The survival curves for subtyping GBM and STAD using the gene expression data.** (**A**) In different cancer types, the expression data of genes those are known tumor-specific drivers with mutation frequency ≥2% were used in subtyping patients. (**B-D**) Similar analysis based on expression data of genes which are known tumor-specific drivers with mutation frequency <2%, predicted driver genes with mutation frequency ≥2% and predicted driver genes with mutation frequency <2% respectively. Different subtypes (S1, S2,...) are indicated by different colored lines.
(TIF)

**S10 Fig. Distribution of the number of predicted personalized driver genes in TARGET and DGIdb.** (**A**) For cancer type GBM, the first pie chart shows the distribution of the number of predicted personalized driver genes in TARGET. Restricted to predicted personalized drivers predicted by PDRWH, there are 21.33% of patients with not less than three actionable driver genes. The second pie chart shows the distribution of the number of predicted personalized driver genes in DGIdb. There are 64.67% of patients with not less than three druggable personalized drivers. The third pie chart is the distribution of the number of predicted personalized driver genes in the union of the two sets. (**B**) The similar pie chart display for cancer type STAD.
(TIF)

**S1 Table. The list of sample numbers and mutant gene numbers in 16 cancer datasets.**
(XLSX)

**S2 Table. Accuracy for the top i-th driver genes predicted by PDRWH in 16 cancer datasets.**
(XLSX)

**S3 Table. The tumor-specific driver genes identified by PDRWH and other methods for five cancer types.**
(XLSX)

**S4 Table. The list of personalized drivers predicted by PDRWH used in stratifying tumor samples and the pathway enrichment analysis.**
(XLSX)

**S5 Table. The list of known driver genes, actionable genes, and druggable genes.**
(XLSX)

## Acknowledgments

We acknowledge the Cancer Genome Atlas Research Network for providing publicly accessible data. The findings presented here are partially or entirely based on data produced by the TCGA Research Network: https://www.cancer.gov/tcga.

## Author Contributions

**Conceptualization:** Naiqian Zhang, Dong Guo, Chenye Wang, Yusen Zhang, Xiaoqi Zheng, Mingyi Wang.

**Data curation:** Yuxuan Pang.

**Formal analysis:** Naiqian Zhang, Fubin Ma, Yuxuan Pang.

**Investigation:** Xiaoqi Zheng.

**Methodology:** Fubin Ma, Chenye Wang.

**Project administration:** Naiqian Zhang, Mingyi Wang.

**Resources:** Yusen Zhang, Xiaoqi Zheng.

**Software:** Yuxuan Pang, Chenye Wang.

**Validation:** Dong Guo.

**Writing – original draft:** Naiqian Zhang, Fubin Ma.

**Writing – review & editing:** Naiqian Zhang, Fubin Ma.

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
