## [Decision Letter · Decision Letter 0]

28 Jan 2024

Dear nqzhangLab Zhang,

Thank you very much for submitting your manuscript "A novel hypergraph model for identifying and prioritizing personalized drivers in cancer" for consideration at PLOS Computational Biology.

As with all papers reviewed by the journal, your manuscript was reviewed by members of the editorial board and by several independent reviewers. In light of the reviews (below this email), we would like to invite the resubmission of a significantly-revised version that takes into account the reviewers' comments. The most substantive comments include (1) lack of clear explanation of the novelty of the approach compared to several other methods that do include various aspects of the proposed method. (2) lack of clarity in the method description, terminologies, parameter choices, etc. (3) a previously proposed superior evaluation criteria, and (4) more independent validations.

We cannot make any decision about publication until we have seen the revised manuscript and your response to the reviewers' comments. Your revised manuscript is also likely to be sent to reviewers for further evaluation.

Sincerely,

Sridhar Hannenhalli

Guest Editor

PLOS Computational Biology

Mark Alber

Section Editor

PLOS Computational Biology

Reviewer's Responses to Questions

**Comments to the Authors:**

Reviewer #1: The authors proposed a method for identifying and prioritizing personalized cancer driver genes. Their method is capable of discovering high-frequency as well as rare potential drivers. In general, the method is novel and interesting, and the algorithm performs well. However, we request some revisions to strengthen the manuscript before we can support publication.

Major:

1) In the section of the introduction, the authors should clarify why the hypergraph works. What's the biological significance? It will be more complete if the authors provide a brief motivation for using a hypergraph model instead of a graph, such as the limitations of graph models for genes or the intuitive advantages of the hypergraph model.

2) Meanwhile, the author should at least provide some references for the hypergraph since its definition is less intuitive than the normal graphs.

For example, the author could include the following references.

[1] Bretto, A. (2013). Hypergraph theory. An introduction. Mathematical Engineering. Cham: Springer.

[2] Zhang, S., Ding, Z., & Cui, S. (2019). Introducing hypergraph signal processing: Theoretical foundation and practical applications. IEEE Internet of Things Journal, 7(1), 639-660.

[3] Barbarossa, S., & Tsitsvero, M. (2016, March). An introduction to hypergraph signal processing. In 2016 IEEE International Conference on Acoustics, Speech and Signal Processing (ICASSP) (pp. 6425-6429). IEEE.

3) In the section “Pre-processing”, a gene whose absolute z-score is more than 2 can be selected as an abnormally expressed gene in the sample. This seems to imply that outlying genes are extracted from the population based on their expression values. Are different thresholds considered in this step?

4) Why does it work better than other algorithms, and what steps work?

5) The authors should clarify whether a directed or undirected PPI network is used.

Minor:

1) In Fig 4b, “%” should be added after the values in parentheses. For example, “33 (84.62%)”.

2) In line 262, the “Supplementary File 2” should be replaced with “S2 File”, to be consistent with the context.

3) The language needs to be improved as some grammatical and spelling errors exist, e.g.,

a) Paper 17 line 259, “is applied to” -> “was applied to”

b) Paper 21 line 290, “and negatively associates” -> “and negatively associate”

c) Page 23 line 324, “to discover” -> “for discovering”

d) Page 29 line 445 “The degree of known driver” -> “The degree of the known driver”

e) Page 31 line 481 “the expression data of genes those are known tumor specific drivers…” is not a complete sentence; “tumor specific” -> “tumor-specific”

Reviewer #2: The authors propose a computational method named PDRWH to identify personalized cancer drivers. This is an important problem in cancer genetics as accurate identification of personalized drivers is critical in determining patient-specific therapies. PDRWH ranks mutated genes based on their impact on differentially expressed genes. It also takes into account other patients when building the personalized driver gene list of a patient. PDRWH has been shown to perform better than personalized driver identification methods as well as cohort-level driver identification methods. The authors have also performed additional evaluation steps where they show that PDRWH can identify rare drivers and gene expression of these rare drivers can stratify the patients into different survival groups. Lastly, they provide experimental results that provide support for driver potential of one of the identified genes, LRP1.

I find the novelty of PDRWH questionable due to the following reasons:

Ranking genes based on their impact on abnormal gene expression is common in the related literature . It has been used in several driver identification methods starting from DriverNet and followed by others e.g. Prodigy, PersonaDrive etc. PersonaDrive is a personalized driver identification method that uses not only the patient of interest but also other patients, so using other patients’ info is also not specific to PDRWH. Many of the driver identification methods that are being compared in this manuscript use random walk strategy. To summarize, PDRWH’s novelty mainly lies in how this random walk is performed in a hyper graph formed by the mutated and “abnormally” expressed genes of each patient. The authors should clarify the contributions of PDRWH more accurately in the Introduction section.

Other major issues:

The authors use Prodigy’s evaluation scheme in Fig 2. PersonaDrive paper lists some disadvantages of this evaluation scheme and propose a modified version. It would be useful to see PDRWH’s performance with this modified evaluation scheme.

I think the evaluations should include two baseline driver identification methods that ranks the mutated genes of a patient

a) based on their degree in the interaction network

b) based on the number of patients that this gene is mutated in

PDRWH determines neighbors of a patient via the concept of shared mutated genes. The authors should include a table with statistics on the number of neighbors for each patient as I would expect a very dense graph using this definition.

Minor issues:

It would be helpful to provide a toy hyper graph in “”Construction of personalized hyper graph” to better explain how weights are assigned to vertices and edges.

The authors should explain how this is achieved “In this case, the walker will be more likely to stabilize at the vertex showing a higher degree of interaction not only in the subnetwork of the target sample but also in those of its neighbors ”

I don’t understand why “potential” is included in the section “PDRWH efficiently identifies both potential frequent and rare drivers “ 

It’s not clear how k is determined in k-means algorithm for tumor stratification analysis.

I couldn’t find for which cancer types LRP1 is identified as a driver. Also, whether this gene is found only by PDRWH should be clarified.

English of the manuscript is problematic and has to be improved significantly. Listing only a few:

-…sample tend to adjacent more…

-….Since all the mutant genes in the target patient should be supposed ..

-Therefore, the weight of hyperedge should be an increased function with their correlation to target patient.

-we got no satisfactory results

-on the decline at the mRNA and protein levels with transfection of three siRNAs

Reviewer #3: This paper introduces a computational method tailored for the personalized identification of cancer driver genes, with its foundation grounded in two critical observations:

- Driver alterations typically induce substantial transcriptional changes in genes located upstream or downstream of signaling pathways. In this context, both the driver gene and the associated upstream and downstream genes closely interact with one another.

- Patients sharing comparable tumor transcriptional profiles exhibit analogous tumor progression, suggesting the implication of shared driver genes in their respective cases.

While the overall concept is innovative and founded on robust biological observations, the paper's clarity is compromised. The lack of essential conceptual definitions, rationale behind specific input choices, and explanations for chosen validation methods contribute to the difficulty in comprehension. Additionally, numerous instances of ambiguous grammar further hinder understanding. Addressing these issues would significantly enhance the paper's accessibility and impact. See comments below:

1) How were mutated genes defined?

a. Was it based on any type of mutation or just non-synonymous mutations?

b. Were focal copy number alterations also considered?

c. What was the rationale behind these decisions?

2) To facilitate personalized identification of cancer drivers, the authors propose a hypergraph model, wherein each patient represents a hyperedge connecting two or more mutated/upregulated/downregulated genes in that patient. Each hyperedge in the hypergraph model gets assigned a weight, which is a function of the correlation between the expression profile of a target patient and expression profiles of its neighbors.

a. A clear definition of how the correlation coefficient was computed is missing.

b. A clear definition of the neighbors of a target patient is missing.

c. Basic intuition suggests that the weights assigned to each hyper edge in the hypergraph should be unique for each target patient. However, the equations defined in the paper seem to suggest a fixed set weights for all hyper edges. This is very counter intuitive considering that the authors claim to predict personalized drivers.

3) The section describing the Generalized Random Walks is written in a highly convoluted and incoherent language that may make it difficult to follow for other readers.

4) For validation, the authors test how well-known cancer type-specific drivers are enriched among the top-ranked driver genes predicted by their method and benchmark against other methods. However, some additional clarifications are needed on how the experiments was performed.

a. What was the rationale behind the choice of only 5 cancer types from TCGA: BRCA, KIRC, LIHC, GBM, STAD? What about other cancer types?

b. Was their proposed method evaluated on genomic and transcriptomic data of patient samples in a pan cancer fashion? Or were the validation experiments run on each cancer type separately? It would be valuable to assess whether their method can successfully identify cancer type-specific drivers within relevant patient populations when evaluated across various cancer types or subtypes. Such a comprehensive pan-cancer evaluation would serve as a definitive proof of concept, demonstrating the method's superiority to predict personalized drivers, specifically tailored to specific patient populations.

c. The study lacks validation on independent datasets such as METABRIC, which houses mutation, copy number and gene expression of 2000 breast cancer patients (https://www.ncbi.nlm.nih.gov/pmc/articles/PMC3440846/)

**Have the authors made all data and (if applicable) computational code underlying the findings in their manuscript fully available?**

Reviewer #1: None

Reviewer #2: Yes

Reviewer #3: Yes

PLOS authors have the option to publish the peer review history of their article (what does this mean?). If published, this will include your full peer review and any attached files.

Reviewer #1: No

Reviewer #2: No

Reviewer #3: No

Figure Files:

Data Requirements:

Please note that, as a condition of publication, PLOS' data policy requir

---

## [Decision Letter · Decision Letter 1]

9 Apr 2024

Dear nqzhangLab Zhang,

We are pleased to inform you that your manuscript 'A novel hypergraph model for identifying and prioritizing personalized drivers in cancer' has been provisionally accepted for publication in PLOS Computational Biology.

However, Reviewer #2 has remaining some minor comments that you should address before finalizing the manuscript.

Best regards,

Sridhar Hannenhalli

Guest Editor

PLOS Computational Biology

Mark Alber

Section Editor

PLOS Computational Biology

Reviewer's Responses to Questions

**Comments to the Authors:**

Reviewer #1: It is revised well

Reviewer #2: All of my comments were addressed sufficiently. Some minor issues:

-The authors should add a discussion on the extremely the low performance in SKCM illustrating the reasons of this.

-We implement a restart random walk -> We implement a random walk with restart

-Fig 1 legend part C and E is too long. Those descriptions should be in the text not in the legend.

Reviewer #3: The authors have carefully addressed previously raised concerns and significantly improved quality of writing, making the manuscript easier to understand.

**Have the authors made all data and (if applicable) computational code underlying the findings in their manuscript fully available?**

Reviewer #1: None

Reviewer #2: Yes

Reviewer #3: Yes

PLOS authors have the option to publish the peer review history of their article (what does this mean?). If published, this will include your full peer review and any attached files.

Reviewer #1: No

Reviewer #2: No

Reviewer #3: No

---

## [Editor Report · Acceptance letter]

22 Apr 2024

PCOMPBIOL-D-23-01883R1 

A novel hypergraph model for identifying and prioritizing personalized drivers in cancer

Dear Dr Zhang,

I am pleased to inform you that your manuscript has been formally accepted for publication in PLOS Computational Biology. Your manuscript is now with our production department and you will be notified of the publication date in due course.

With kind regards,

Anita Estes
